# Hydroxytyrosol Reprograms the Tumor Microenvironment in 3D Melanoma Models by Suppressing ERBB Family and Kinase Pathways

**DOI:** 10.3390/ijms26146957

**Published:** 2025-07-20

**Authors:** David Tovar-Parra, Marion Zammit Mangion

**Affiliations:** 1Department of Physiology and Biochemistry, Faculty of Medicine and Surgery, University of Malta, MSD 2080 Msida, Malta; david.tovar@inrs.ca; 2Institut National de la Recherche Scientifique INRS, Centre Armand-Frappier Santé Biotechnologie, Laval, QC H7V 1B7, Canada; 3Centre for Molecular Medicine and Biobanking, Department of Physiology and Biochemistry, University of Malta, MSD 2080 Msida, Malta

**Keywords:** melanoma, hydroxytyrosol, melanocyte, proteomic, spheroid

## Abstract

Malignant cutaneous melanoma is among the most aggressive forms of skin cancer, characterized by high metastatic potential and frequent resistance to standard therapies. Hydroxytyrosol, a phenolic compound derived from extra virgin olive oil, has shown promising anticancer properties in various models, yet its effects in 3D melanoma systems remain poorly understood. In this study, we used paired 3D spheroid models of non-tumorigenic (HEMa) and melanoma (C32) to assess the therapeutic potential of hydroxytyrosol. To evaluate the anti-tumoral effect of hydroxytyrosol, we performed cytotoxicity, metastasis, invasiveness, cell cycle arrest, apoptotic, and proteomic assays. Hydroxytyrosol treatment significantly impaired spheroid growth, reduced cell viability, and induced cell cycle arrest and apoptosis in C32 spheroids, with minimal cytotoxicity observed in HEMa models. Proteomic profiling further demonstrated that hydroxytyrosol selectively downregulated a network of oncogenic proteins, including ERBB2, ERBB3, ERBB4, VEGFR-2, and WIF-1, along with suppression of downstream PI3K-Akt and MAPK/ERK signaling pathways. In conclusion, compared to dabrafenib, hydroxytyrosol exerted a broader range of molecular effects and was more selective toward tumor cells. These findings support the use of hydroxytyrosol as a multi-targeted agent capable of attenuating melanoma progression through suppression of kinase signaling and tumor-stromal interactions.

## 1. Introduction

Malignant cutaneous melanoma represents a significant proportion of skin cancers worldwide due to its aggressive behavior and metastatic potential [1]. According to the International Agency for Research on Cancer, by 2025, around 354,000 new cases of melanoma and 62,600 related deaths are projected [2]. Although targeted therapies (BRAF/MEK inhibitors) and immune checkpoint inhibitors have improved outcomes, intrinsic and acquired resistance, often driven by reactivation of mitogen-activated protein kinase (MAPK) and phosphatidylinositol 3-kinase (PI3K)/Akt pathways, remains a significant challenge in the clinical management of melanoma [3]. Metastatic melanoma continues to pose therapeutic challenges and remains largely incurable [3,4].

Immunotherapies, including immune checkpoint inhibitors such as anti-PD-1 and anti-CTLA-4, have transformed melanoma care and produce durable responses in 30–40% of patients [5,6,7]. Similarly, targeted therapies against BRAF and MEK kinase have shown efficacy in patients harboring specific genetic mutations such as BRAF V600E and others, offering improved progression-free survival compared to chemotherapy [8,9,10]. Nevertheless, most patients eventually relapse due to primary or acquired resistance mechanisms, often involving the reactivation of MAPK or PI3K-Akt signaling, the upregulation of receptor tyrosine kinases (RTKs), or immune evasion [11,12]. In clinical settings where immunotherapy or targeted therapies fail, treatment options are limited, and cytotoxic chemotherapy remains the fallback in many cases [13,14,15]. Thus, the identification of complementary or alternative therapeutic agents, particularly those that can target proliferative signaling and invasive phenotypes, remains a high priority.

Natural compounds derived from plants have gained considerable attention in cancer research due to their diverse pharmacological properties and potential as anticancer agents [16]. Among these compounds, phenolic compounds such as hydroxytyrosol (HT), tyrosol, oleacein, and oleuropein, found in extra virgin olive oil (EVOO) [17,18,19], have emerged as promising candidates for cancer treatment and also for their diverse pharmacological properties, including antioxidant, anti-inflammatory, and anti-cancer effects [20,21,22]. Phenolic compounds such as oleuropein reduce nuclear factor kappa B (NF-kB) expression in different tumors, leading to increased apoptosis in breast cancer cell lines and decreased hypoxia-inducible factor-1 (HIF-1) expression in human colorectal cancer cells [23,24]. Hydroxytyrosol, a potent antioxidant and the main bioactive component of EVOO, has received considerable attention for its antiproliferative, anti-inflammatory, and pro-apoptotic properties [23]. Studies in glioblastoma, neuroblastoma, and hepatocellular carcinoma models have reported that HT suppresses cell viability, impairs migration, enhances ROS production, and modulates apoptotic signaling pathways [25]. In melanoma, HT has been shown to inhibit proliferation and induce DNA damage via p53 and γH2AX activation while concurrently suppressing AKT phosphorylation [26]. However, these studies have largely relied on traditional two-dimensional (2D) culture systems, which fail to capture the structural and biochemical complexity of the tumor microenvironment.

Three-dimensional (3D) spheroid models offer a more physiologically relevant platform for evaluating drug responses by mimicking the spatial architecture, cell–cell interactions, and diffusion gradients present in tumors [27,28]. To date, the effects of hydroxytyrosol on 3D melanoma models, particularly its ability to modulate receptor tyrosine kinases, erb-b receptor tyrosine kinase (ERBB) family members, and EMT-related pathways, remain poorly understood. In this study, we applied paired 3D spheroid systems derived from non-tumorigenic melanocytes (HEMa) and amelanotic melanoma cells (C32) to investigate the effects of HT on cell viability, migration, invasion, EMT marker expression, cell cycle progression, and proteomic signaling networks. Our results reveal that hydroxytyrosol selectively disrupts key oncogenic pathways, including ERBB2/3/4, PI3K-Akt, and MAPK signaling, providing new insights into its anti-melanoma potential and highlighting its utility as a candidate for adjunct therapy in melanoma.

## 2. Results

### 2.1. Hydroxytyrosol Reduces Spheroid Growth from Tumoral and Non-Tumoral Cells

Both non-tumorigenic HEMa and tumorigenic C32 melanoma spheroids were treated from day 5 to day 11 with increasing concentrations of hydroxytyrosol (HT; 0–100 ppm). In non-tumor HEMa spheroids, treatment with 25 ppm of HT had minimal effect on spheroid morphology or size, which remained comparable to untreated controls. However, higher concentrations (50 and 100 ppm) resulted in a progressive reduction in spheroid area, with shrinkage ranging from 10% to 30% relative to baseline (Figure 1A,B). In contrast, tumor C32 spheroids exhibited a more pronounced response to HT. While 25 ppm HT did not significantly alter spheroid size, treatment with 50 and 100 ppm induced a clear and statistically significant reduction in spheroid area compared to controls, indicative of a dose-dependent growth inhibitory effect (Figure 1A,B).

To further quantify the cytotoxicity of HT in both cell types, a luminescent 3D viability assay was used to calculate half-maximal inhibitory concentrations (IC_50_) following 72 h of exposure to escalating doses of HT and dabrafenib. In non-tumor HEMa spheroids, the IC_50_ value for HT was 105.7 ppm, whereas dabrafenib demonstrated greater potency with an IC_50_ of 24.5 ppm. In contrast, tumor C32 spheroids exhibited significantly higher sensitivity to both compounds, with an IC_50_ of 51.1 ppm for HT and 10.9 ppm for dabrafenib (Figure 1C). These findings suggest that C32 melanoma spheroids are more susceptible to HT-induced growth inhibition than non-tumorigenic HEMa spheroids, and that HT exerts a dose-dependent cytotoxic effect similar in profile to the clinically used BRAF inhibitor dabrafenib.

### 2.2. Hydroxytyrosol Reduces Spheroid Migration on Matrigel

To determine whether hydroxytyrosol affects the migratory behavior of melanoma spheroids, HEMa and C32 spheroids were embedded in Matrigel and monitored over 5 days. Figure 2A shows that vehicle-treated HEMa spheroids progressively expanded outward, reaching approximately 120–130% of their initial area by Day 5 (a 20–30% increase in spheroid outgrowth area). In contrast, HEMa spheroids treated with HT (100 ppm) remained compact and even contracted slightly over time, shrinking to about 80–85% of their initial area by Day 5 (a 15–20% decrease). These results indicate that HT treatment markedly suppresses the collective migration and spreading of HEMa spheroids on a reconstituted basement membrane, comparable to the effect of dabrafenib (Figure 2A).

A similar differential migration pattern was observed in tumor C32 melanoma spheroids (Figure 2B). Untreated C32 spheroids exhibited aggressive outgrowth on Matrigel, with the projected area expanding to roughly 160–180% of the initial spheroid size by Day 5 (a 60–80% increase). HT treatment (50 ppm) only modestly affected C32 spheroid expansion, as the final area remained comparable to vehicle controls (a 60–80% increase). In contrast, dabrafenib (10 ppm) dramatically inhibited C32 spheroid migration: Dabrafenib-treated spheroids failed to expand and instead became significantly smaller, with final areas approximately 50–70% lower than those of control spheroids by Day 5. Thus, while HT effectively constrained spheroid migration in the HEMa model, its impact on the highly aggressive C32 spheroids was less pronounced, whereas dabrafenib robustly suppressed spheroid outgrowth in both cell lines (Figure 2B).

### 2.3. Hydroxytyrosol Inhibits Invasive Capacity in Both Tumor and Non-Tumor Spheroids

We next evaluated the invasive capacity of spheroids using a Boyden chamber Matrigel invasion assay (Figure 3B). In non-tumor HEMa spheroids, HT (100 ppm) did not significantly change the number of cells invading through the Matrigel-coated membrane compared to the vehicle-treated control (invasion remained ~100% of control levels). Dabrafenib (25 ppm), however, caused a modest but significant decrease in HEMa invasion by approximately 10–15% relative to the control (*p* < 0.05). In the more aggressive C32 spheroids, HT (50 ppm) produced a clear reduction in invasive potential: Treated C32 spheroids showed 35–40% fewer invading cells than controls (*p* < 0.05). This inhibitory effect was comparable to that of dabrafenib (10 ppm), which reduced C32 invasion by about 42% relative to the control (* *p* < 0.001). These data indicate that HT can impair the invasive behavior of melanoma spheroids, significantly so in the C32 cell line (Figure 3B).

### 2.4. Hydroxytyrosol Downregulates Mesenchymal EMT Markers

Next, we determined whether hydroxytyrosol influences the expression of EMT-related markers, such as vimentin and N-cadherin, in spheroids. Flow cytometry was used to quantify the expression of vimentin and N-cadherin in dissociated spheroid cells at 1 and 3 days post-treatment (Figure 3C–E). In non-tumor HEMa spheroids, HT (100 ppm) treatment led to a pronounced decrease in EMT marker-positive cells at both time points. Specifically, the proportion of cells expressing vimentin or N-cadherin was reduced by approximately 25–35% in HT-treated spheroids compared to controls, at both Day 1 and Day 3. This degree of EMT marker downregulation was comparable to that achieved with dabrafenib (25 ppm) in HEMa spheroids, where vimentin and N-cadherin levels were similarly lower than those of controls (Figure 3D).

In tumor C32 spheroids, HT (50 ppm) had a minimal immediate effect on vimentin and N-cadherin expression at Day 1. By Day 3, however, HT-treated C32 spheroids showed a measurable reduction (~15–20%) in the vimentin-positive and N-cadherin-positive cell populations relative to untreated controls. In comparison, dabrafenib (10 ppm) induced an earlier and more sustained suppression of these EMT markers in C32 spheroids: Already, at Day 1, dabrafenib-treated C32 cells displayed lower vimentin and N-cadherin levels than controls, and this downregulation persisted through Day 3 (Figure 3E). These results demonstrate that HT can attenuate mesenchymal marker expression in melanoma spheroids, mirroring the EMT-inhibitory effects of dabrafenib, although the onset and magnitude of the response vary between the two melanoma cell lines.

### 2.5. Hydroxytyrosol Modulates Cell Cycle Distribution in Melanoma Spheroids

In non-tumor HEMa spheroids, hydroxytyrosol treatment did not markedly alter the cell cycle profile at Day 1 compared to untreated controls (Figure 4A). By Day 3, however, HT-exposed HEMa spheroids showed a slight decrease in the G1-phase population (approximately 10% lower) and in the S-phase population (around 5% lower) relative to the control (Figure 4C). The BRAF inhibitor dabrafenib produced minimal changes in HEMa cell cycle distribution at Day 1. By Day 3, dabrafenib-treated HEMa spheroids exhibited a modest reduction in the fractions of S and G2 phase cells (S phase decreased from ~13% in controls to ~8%; G2 from ~8% to ~5%) with no substantial change in G1 (Figure 4C).

In the tumor C32 melanoma spheroids, HT likewise had no significant effect on cell cycle phase distribution at Day 1 (Figure 4B). After 3 days of treatment, HT-treated C32 spheroids showed a reduced G1-phase fraction (approximately 12% below the control) accompanied by an increase in the sub-G1 population by about 20% (indicating more sub-diploid/apoptotic cells) relative to the control (Figure 4D). Dabrafenib treatment of C32 spheroids led to an increased G1 fraction at Day 1 (rising from ~53% in control to ~69% with dabrafenib; Figure 4B). By Day 3, dabrafenib had caused broad declines in the G1, S, and G2 populations (roughly 17%, 6%, and 5% lower, respectively, than in controls), alongside a pronounced accumulation of cells in sub-G1 (~30% higher than control; Figure 4D).

Taken together, these results indicate that hydroxytyrosol exerts mild cell cycle-modulating effects in HEMa spheroids and promotes sub-G1 accumulation in C32 spheroids over time, consistent with an emerging anti-proliferative effect in tumor cells.

### 2.6. Hydroxytyrosol Induces Selective Cell Death in Melanoma and Non-Tumor Spheroids

In the non-tumor HEMa spheroids, hydroxytyrosol induced a substantial increase in cell death markers. At Day 1, HT-treated HEMa spheroids exhibited slightly higher levels of necrotic cells (~9% increase) and late apoptotic/necrotic cells (~3% increase) compared to untreated controls (Figure 5A,C). By Day 3, the early apoptotic cell fraction in HEMa spheroids had risen from roughly 1% in control conditions to about 9% under hydroxytyrosol treatment (Figure 5C). In comparison, dabrafenib caused a small amount of cell death in HEMa spheroids by Day 1 (~8% of cells non-viable), which increased to approximately 25% non-viability by Day 3 (Figure 5A,C).

In the melanoma C32 spheroids, hydroxytyrosol triggered only a minor increase level of cell death. HT treatment increased the overall non-viable cell population in C32 spheroids at both Day 1 and Day 3 relative to the control group (Figure 5B,D). However, dabrafenib exposure in C32 spheroids resulted in about 19% of the cells being non-viable by Day 3 (Figure 5B,D). These findings suggest that hydroxytyrosol induces cell death in non-tumor melanocytic spheroids.

### 2.7. Hydroxytyrosol Downregulates Pro-Tumorigenic Protein Networks in Hema Spheroids

To assess the impact of hydroxytyrosol (HT) on the proteomic profile of non-tumorigenic spheroids, a targeted protein analysis was performed using the Olink Target 96 Oncology II panel. Unsupervised hierarchical clustering of protein expression in HEMa spheroids revealed three distinct expression patterns, suggesting biologically meaningful segmentation between experimental conditions (Figure 6A). Differential expression analysis showed that HT treatment led to the downregulation of 25 proteins and upregulation of 2 proteins compared to vehicle-treated controls (Figure 6B), indicating a substantial shift in the proteomic landscape. Among the most significantly downregulated targets were integrin Subunit Beta 5 (ITGB5) (−1.89 NPX, *p* = 7.3 × 10^−6^), Vascular endothelial growth factor receptor 2 (VEGFR-2) (−1.25, *p* = 7.1 × 10^−5^), Wnt inhibitory factor 1 (WIF-1) (−1.22, *p* = 6.5 × 10^−5^), transforming growth factor alpha (TGF-α) (−1.50, *p* = 4.1 × 10^−4^), and ERBB2/3 (−1.15/−1.17, *p* < 0.002) proteins closely linked to angiogenesis, adhesion, and proliferative signaling.

Gene Ontology (GO) enrichment analysis of the downregulated proteins revealed involvement in biological processes such as regulation of angiogenesis, positive regulation of kinase activity, and endothelial cell function (Figure 6C). Molecular function categories included growth factor binding, extracellular matrix (ECM) interactions, cadherin binding, and transmembrane receptor protein kinase activity. Pathway analysis using KEGG indicated significant enrichment in downregulated proteins associated with MAPK signaling, EGFR tyrosine kinase inhibitor resistance, and PI3K-Akt signaling—key oncogenic pathways commonly implicated in melanoma progression (Figure 6D).

These findings suggest that HT broadly attenuates pro-tumorigenic signaling networks in non-malignant melanocytic spheroids by downregulating key proteins involved in angiogenesis, cell proliferation, and ECM interactions. Taken together, these results indicate that hydroxytyrosol significantly alters the expression of proteins related to angiogenesis and oncogenic signaling in non-tumorigenic spheroids, potentially contributing to a less permissive environment for tumor initiation.

### 2.8. Hydroxytyrosol Downregulates Pro-Tumorigenic Protein Networks in C32 Spheroids

To investigate the proteomic impact of hydroxytyrosol on tumor spheroids, C32 melanoma spheroids were analyzed using the same targeted panel. Unlike HEMa spheroids, unsupervised clustering of C32 samples did not yield clearly distinct groupings according to treatment condition, suggesting greater inter-sample variability or a more subtle treatment effect (Figure 7A). Nonetheless, volcano plot analysis revealed that 11 proteins were significantly downregulated in HT-treated C32 spheroids compared to controls, with no proteins significantly upregulated (Figure 7B). Among these, several proteins are associated with key oncogenic and immune regulatory pathways, including ERBB4 (–1.33 NPX, *p* = 1.5 × 10^−4^), CYR61 (–1.01, *p* = 0.0069), CDKN1A (p21, –0.83, *p* = 0.0215), FADD (–0.86, *p* = 0.0436), and ABL1 (–0.77, *p* = 0.0383). Other significantly reduced proteins include TRAIL, TGFR-2, ANXA1, SPARC, FURIN, GPNMB, and ERBB2, all of which play roles in melanoma progression, immune modulation, or resistance to targeted therapies [29,30].

GO enrichment analysis of the downregulated proteins identified associations with biological processes such as the ERK1/2 signaling cascade, the regulation of immune responses, and cellular stress pathways (Figure 7C). Affected molecular functions included protein tyrosine kinase activity and growth factor binding. KEGG pathway enrichment analysis revealed involvement in oncogenic pathways such as proteoglycans in cancer, PI3K-Akt signaling, and ErbB signaling (Figure 7C,D). Notably, downregulated targets such as ERBB4, ERBB2, and ABL1 directly interface with these pathways, while FADD and CDKN1A are associated with apoptosis regulation and cell cycle arrest. Together, these findings show that hydroxytyrosol modulates key signaling proteins in melanoma spheroids, attenuating oncogenic and immune-modulatory networks relevant to tumor progression and therapy resistance.

## 3. Discussion

In this study, we investigated the anti-tumor effects and therapeutic potential of hydroxytyrosol (HT), a phenolic compound derived from extra virgin olive oil, in 3D melanoma spheroid models. Using paired non-tumorigenic HEMa and tumorigenic C32 spheroids, we demonstrate that HT selectively impairs tumor cell viability, proliferation, migration, invasion, and EMT marker expression. In addition, HT promotes cell cycle arrest, induces apoptosis, and downregulates key receptor tyrosine kinases and kinase effectors involved in melanoma progression.

Compared to dabrafenib, a clinically approved BRAF inhibitor, HT exhibited comparable anti-proliferative and anti-invasive effects in C32 melanoma spheroids. Notably, HT exerted minimal cytotoxicity in non-tumor HEMa spheroids, suggesting a degree of tumor selectivity. This is particularly relevant given the limitations of targeted therapies, including toxicity and acquired resistance, which often arise from feedback activation of alternative pathways such as PI3K-Akt and MAPK [15,31]. While dabrafenib directly inhibits BRAF activity [32], HT induced a broader suppression of oncogenic proteins involved in tumor-stroma crosstalk and angiogenesis, including VEGFR-2, WIF-1, ITGB5, and ERBB2/3/4. Lamy et al., reported that HT inhibits in vitro angiogenesis in endothelial cells by autophosphorylation sites of VEGFR-2 (Tyr951, Tyr1059, Tyr1175, and Tyr1214) [33].

Melanoma is classically driven by mutations in genes such as BRAF, NRAS, and c-KIT, which promote uncontrolled cell growth and survival [34]. Our findings reveal that HT downregulates ERBB2, ERBB3, and ERBB4 receptors, alongside downstream suppression of PI3K-Akt and MAPK/ERK pathways. These observations align with previous reports highlighting the role of ERBB family members in promoting melanoma invasiveness, metastasis, and therapy resistance [35,36]. In particular, ERBB3 is overexpressed in advanced melanoma and contributes to drug resistance and EMT-like phenotypes [36]. By disrupting ERBB signaling, HT likely attenuates the activation of RAS/RAF/MEK/ERK and PI3K/Akt cascades, central regulators of cell cycle progression and survival [37,38]. This molecular inhibition was reflected in our spheroids as decreased ERK signaling, G1/S arrest, and sub-G1 accumulation, consistent with increased apoptotic activity. These findings are in line with studies showing that HT reduces AKT levels and induces γH2AX-mediated DNA damage in melanoma monolayers [39]. Similarly, HT has been shown to inactivate AKT and NF-κB, leading to a reduced expression of Cyclin D1, c-Myc, and Bcl-2 in hepatocellular carcinoma models [40]. In our spheroids, we observed reduced levels of these proliferative and anti-apoptotic markers, further supporting the role of HT in suppressing tumor-promoting pathways.

In addition to affecting proliferation and survival, HT modulated cell phenotype by reducing EMT-related proteins such as N-cadherin and vimentin. These molecular changes were accompanied by functional inhibition of spheroid migration and invasion [41]. EMT is a well-recognized mechanism underlying metastatic potential and immune evasion in melanoma [42,43]. Our results suggest that HT promotes a mesenchymal-to-epithelial shift, thereby limiting tumor aggressiveness. In triple-negative breast cancer cells, Cruz-Lozano et al. reported that HT suppressed EMT by downregulating SNAIL, SLUG, ZEB1, and vimentin [44], which supports our findings in melanoma spheroids. Overall, these results underscore the multifaceted role of hydroxytyrosol in disrupting oncogenic signaling, EMT, and invasive behaviors in melanoma. Importantly, this study highlights the relevance of 3D spheroid systems in modeling tumor complexity and evaluating drug responses in a physiologically relevant context.

## 4. Materials and Methods

### 4.1. Cell Culture and Spheroid Formation

The human epidermal melanocyte cell line (HEMa; ATCC PCS-200-013) and the amelanotic melanoma cell line (C32; ATCC CRL-1585) were cultured in Dermal Cell Basal Medium (ATCC PCS-200-030) supplemented with the Adult Melanocyte Growth Kit (ATCC PCS-200-042) were purchased from the American Type Culture Collection (ATCC, Manassas, VA, USA), and in RPMI-1640 medium (Sigma R8758, Waltham, MA, USA) supplemented with 10% fetal bovine serum (FBS, Gibco, Thermo Fisher Scientific, Waltham, MA, USA) and 1% penicillin/streptomycin (Sigma-Aldrich Chemie Gmbh, Wesel, Germany), respectively. All cultures were maintained at 37 °C in a humidified atmosphere containing 5% CO_2_.

To generate spheroids, 10,000 cells were seeded in 100 µL of their respective media into ultra-low attachment (ULA) round-bottom 96-well plates (Thermo Scientific, 174927, Waltham, MA, USA). Media were gently replaced every 3 days by aspirating 80 µL and replenishing with fresh medium to preserve spheroid integrity. Both tumor C32 and non-tumor HEMa spheroids were cultured for 5 days before treatments. Spheroids were then exposed to 3-hydroxytyrosol (Sigma-Aldrich, SGM-H4291-25MG, Chemie Gmbh, Wesel, Germany) at concentrations ranging from 1 to 100 parts per million (ppm). Dabrafenib (MedChemExpress, HY-12057, Wesel, Germany) served as the positive control (BRAF inhibitor) concentration range of 0–100 ppm, and ethanol (EtOH) was used as the vehicle control, diluted to a final concentration of 0.1% (*v/v*). All treatments were performed in biological triplicate unless otherwise stated.

### 4.2. Cell Viability and Dose–Response Assays

Cell viability and half-maximal inhibitory concentrations (IC_50_) were assessed using the CellTiter-Glo^®^ 3D Cell Viability Assay (Promega, G9682, (Madison, WI, USA), Chemie Gmbh, Socorex Isba SA (Ecublens, Switzerland)), according to the manufacturer’s protocol. After 72 h of treatment with HT, dabrafenib, or vehicle control across a concentration range (0–100 ppm), spheroids were transferred to opaque-walled 96-well plates. An equal volume (100 µL) of CellTiter-Glo^®^ 3D reagent was added to each well. Plates were shaken for 5 min and incubated at room temperature for an additional 25 min before luminescence was measured using a Spark 20M microplate reader (TECAN) (Mnnedorf, Switzerland). Dose–response curves and IC_50_ values were calculated using GraphPad Prism 10 (GraphPad Software, San Diego, CA, USA). All results are presented as mean ± standard deviation (SD) from three independent experiments.

### 4.3. Spheroid Growth and Morphometric Analysis

To assess the effects of HT and dabrafenib on spheroid morphology and growth dynamics, treatments were initiated on day 5 and monitored until day 11. Spheroids were imaged at days 5, 7, 9, and 11 using the EVOS™ M7000 Imaging System-2 (Advanced Microscopy Group, Paisley, Scotland, UK). The projected 2D area of each spheroid was measured using ImageJ Fiji software (NIH) version 1.54 (National Institute of Health, Bethesda, MD, USA). Spheroid size over time was analyzed to evaluate treatment-induced growth inhibition or structural disruption in both C32 and HEMa spheroids.

### 4.4. Spheroid Migration Assay

Spheroid migratory capacity was evaluated on Matrigel-coated plates (Corning, 356237, (Corning, NY, USA)) prepared by diluting Matrigel 1:4 in cold culture medium and incubating for 30 min at 37 °C to allow polymerization [10]. Following Matrigel solidification, four spheroids per condition were seeded onto each coated well and treated with the IC_50_ concentration of HT, dabrafenib, or vehicle control. Images were captured at 0, 24, 48, 72, 96, and 120 h using the EVOS™ M7000 system, and the area invaded by migrating cells was quantified using ImageJ version 1.54 (National Institute of Health, Bethesda, MD, USA).

### 4.5. Spheroid Invasion Analysis

The invasion potential of HT treated and control spheroids was assessed using the ECMatrix™ Cell Invasion Assay (Millipore-Sigma, Wesel, Germany) in 24-well plates. Spheroids were generated over 5 days and serum-starved in supplement-free medium for 24 h prior to invasion analysis. ECMatrix inserts were hydrated with 300 µL of serum-free medium for 30 min at 37 °C. After removing excess medium from the upper chamber, 250 µL of treatment medium containing HT (IC_50_), dabrafenib, or vehicle was added to the upper chamber, while 500 µL of complete medium was placed in the lower chamber.

Invasion was monitored every 24 h for 72 h using the EVOS™ XL Core Imaging System-2 (Advanced Microscopy Group, Paisley, Scotland, UK). At the endpoint, non-invading cells and spheroids were removed by transferring inserts to a clean well containing 225 µL of Cell Detachment Solution (Millipore-Sigma) for 30 min at 37 °C. Then, 75 µL of Lysis Buffer containing fluorescent dye was added and incubated for 15 min. A 200 µL aliquot was transferred to a black-walled 96-well plate, and fluorescence (Ex/Em = 480/520 nm) was measured using the TECAN Spark 20M (Mnnedorf, Switzerland).

### 4.6. Flow Cytometric and Phenotypic Markers

To characterize epithelial-to-mesenchymal transition (EMT)-related phenotypes, spheroids were treated with HT (IC_50_), dabrafenib, or vehicle for 24, 48, and 72 h. Ten spheroids per condition were enzymatically dissociated in 0.25% trypsin-EDTA at 37 °C for 5 min, followed by PBS washing. Cells were stained with anti-N-cadherin (Abcam, ab19348, Cambridge, UK) and vimentin–Alexa Fluor 647 (Abcam, ab195878, Cambridge, UK). Secondary antibodies conjugated to Alexa Fluor 568 (Abcam, ab150077, and ab175473, Cambridge, UK) were used at 1:1000 dilution for 30 min at room temperature. Samples were acquired on an Attune™ NxT Flow Cytometer (Attune™ NxT Acoustic Focusing Cytometer, Life Technologies, Carlsbad, CA, USA), and analyzed using FlowJo Version X. The percentage of marker-positive cells was calculated from three independent experiments and reported as mean ± SD.

### 4.7. Cell Cycle Analysis and Apoptosis Assay

For cell cycle and apoptosis analyses, spheroids were pre-starved for 24 h in 0% FBS medium, then treated with HT (IC_50_), dabrafenib, or vehicle control for 24 and 72 h. After incubation, spheroids were dissociated in 0.05% trypsin-EDTA and collected by centrifugation (300× *g*, 5 min). For cell cycle analysis, cells were fixed in 70% cold ethanol for ≥2 h, washed, and stained with propidium iodide (50 µg/mL) and RNase A (100 µg/mL) for 1 h at 37 °C. DNA content was assessed using the Attune™ NxT cytometer, and phase distribution was analyzed with FlowJo VX. For apoptosis, the Apoptosis and Necrosis Assay Kit (Abcam—ab176749, Cambridge, UK) was used. Cells were stained with Apopxin Green, 7-AAD (1:200), and CytoCalcein 450 as per the manufacturer’s protocol. After 60 min incubation, samples were diluted to 300 µL and analyzed on the Attune™ NxT cytometer (Attune™ NxT Acoustic Focusing Cytometer, Life Technologies, Carlsbad, CA, USA). Percentages of viable, apoptotic, and necrotic cells were quantified using FlowJo Version X.

### 4.8. Protein Extraction and Proteomic Profiling

Spheroids were cultured for 5 days and treated with HT (IC_50_), dabrafenib, or vehicle for 24 and 72 h. Ten spheroids per condition were washed with ice-cold PBS and lysed using Bio-Plex Cell Lysis Solution (Bio-Rad, 171304011, LS RTU GmbH, Weiterstadt, Germany) on ice for 20 min with agitation. Lysates were clarified (14,000× *g*, 15 min, 4 °C), and protein concentration was determined using the Pierce BCA Protein Assay Kit (Thermo Fisher Scientific). All samples were normalized to 0.5 mg/mL. Proteomic profiling was conducted using the Olink Target 96 Oncology II panel (Olink Proteomics, Uppsala, Sweden), which employs proximity extension assays (PEA) for simultaneous quantification of 96 cancer-related proteins. All assays were performed in biological triplicate, and negative controls were used to define the assay’s limit of detection (LOD). Data are presented as mean ± SD.

### 4.9. Bioinformatic and Statistical Analysis

For comprehensive data analysis, Normalized protein expression (NPX) values generated by the Olink Target 96 Oncology II panel were exported from Olink NPX Manager (v.2.0; Olink Proteomics, Uppsala, Sweden). All NPX data were log_2_-transformed prior to downstream analyses, such that each unit increase represents a doubling of protein concentration. Group-wise distribution of log_2_-NPX values was first assessed for normality and homoscedasticity using the Shapiro–Wilk and F tests, respectively. Parametric comparisons employed Student’s t-test or one-way ANOVA with Welch’s correction when variances were unequal; non-parametric analyses used Mann–Whitney U or Kruskal–Wallis tests as appropriate. P-values were adjusted for multiple testing by the Benjamini–Hochberg (FDR) method, with FDR  <  0.05 considered significant. Fold-changes were calculated on the linear scale using geometric means of each group and back-transformed to log_2_.

Principal component analysis (PCA) and linear discriminant analysis (LDA) were performed in R (v.4.2.3; R Foundation, Duke University, in Durham, NC, USA) to explore sample clustering and discriminative protein signatures. Volcano plots, heatmaps, bar plots, boxplots, and dot plots were generated using the ggplot2 package to visualize differential regulation patterns. Proteins exhibiting significant differential expression (FDR  <  0.05 and log_2_-fold-change ≥ 1.5) were subjected to Gene Ontology (GO) and Kyoto Encyclopedia of Genes and Genomes (KEGG) pathway enrichment analyses facilitated by the SRplot package version 2 (http://www.bioinformatics.com.cn, accessed 12 April 2025) and R software environment. All statistical tests and graphics (except those described above) were completed in GraphPad Prism 9.0 (GraphPad Software, San Diego, CA, USA). Final figures were assembled and annotated in Inkscape 1.2.2. Quantitative results are presented as mean ± SD from at least three independent experiments.

## 5. Conclusions

Our findings demonstrate that hydroxytyrosol effectively reprograms the tumor microenvironment in 3D melanoma spheroids by downregulating the ERBB2/3/4 axis and suppressing key oncogenic pathways such as PI3K-Akt and MAPK/ERK. These molecular changes resulted in reduc45ed tumor cell proliferation, increased apoptosis, and diminished EMT and invasion. Importantly, HT exhibited tumor-selective activity, with minimal cytotoxic effects on non-tumor melanocytic spheroids. These data suggest that HT holds promise as a multi-targeted therapeutic agent capable of modulating tumor cell behavior and the extracellular milieu in melanoma.

## Figures and Tables

**Figure 1 ijms-26-06957-f001:**
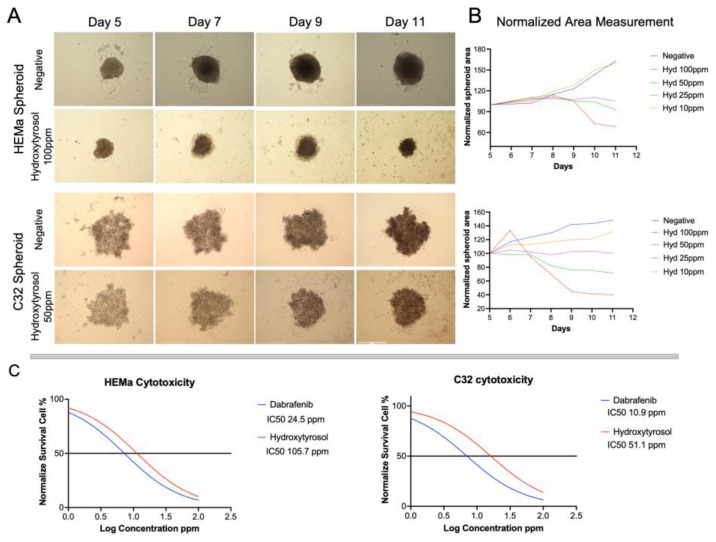
Hydroxytyrosol (HT) reduces spheroid growth in a dose-dependent manner. (**A**) Representative brightfield images of non-tumorigenic HEMa and tumorigenic C32 spheroids treated with HT, dabrafenib, or vehicle. Scale bar = 200 µm. (**B**) Quantification of spheroid projected area over time (days 5, 7, 9, and 11). (**C**) Dose–response curves and calculated IC_50_ values for HT and dabrafenib in HEMa and C32 spheroids, assessed after 72 h using CellTiter-Glo^®^ 3D viability assay. IC_50_ values were derived from non-linear regression models; data shown as mean ± SD from three biological replicates.

**Figure 2 ijms-26-06957-f002:**
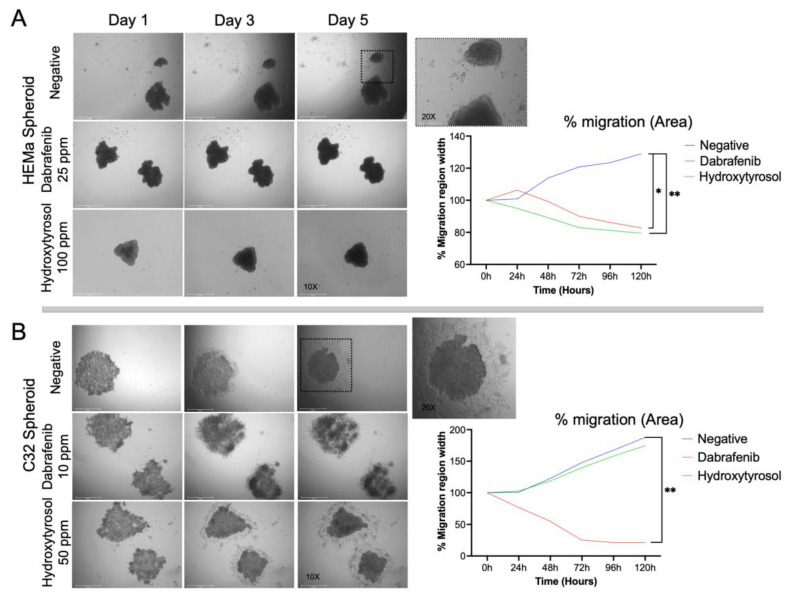
Hydroxytyrosol inhibits spheroid migration on Matrigel. (**A**) Representative images and quantification of HEMa spheroid migration over 5 days in Matrigel-coated wells. (**B**) C32 spheroid migration over the same period. Graphs represent spheroid outgrowth area over time, expressed as a percentage relative to Day 0. Data are shown as mean in % (*n* = 4 spheroids per condition) (* *p* < 0.05, ** *p* < 0.01). See Appendix A for complete statistical analysis.

**Figure 3 ijms-26-06957-f003:**
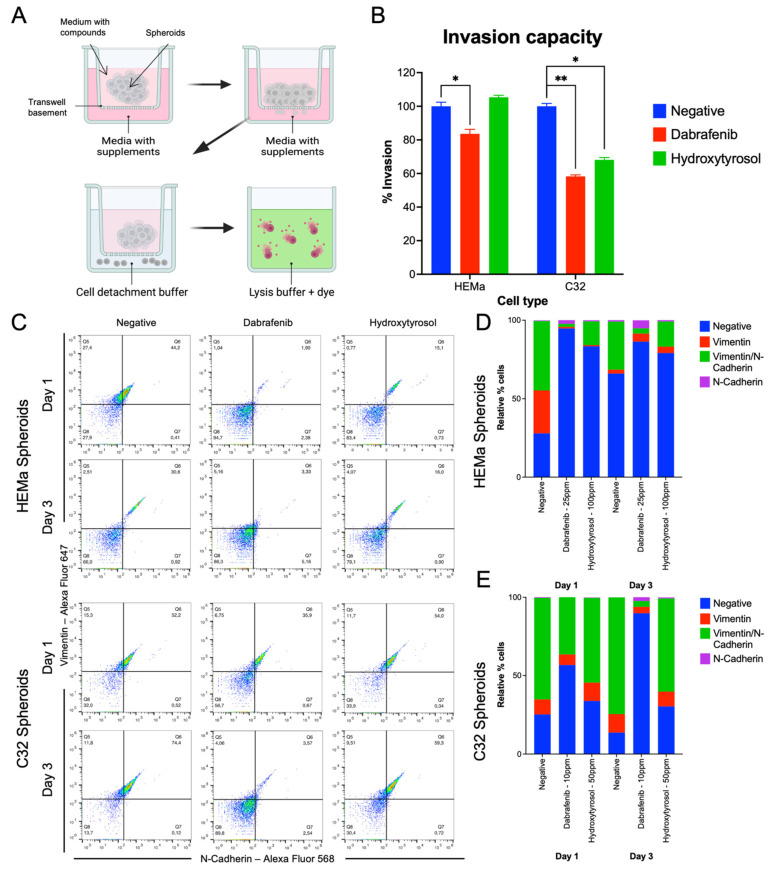
Hydroxytyrosol reduces invasive capacity in C32 spheroids. (**A**) Representative images for invasion analysis. (**B**) Graphical representation of the invasion capacity of HEMa and C32. (**C**–**E**) Flow cytometry analysis of EMT-related markers Vimentin (Q5), N-cadherin (Q7), and Vimentin/N-Cadherin (Q6) in HEMa (**C**,**D**) and C32 (**C**,**E**) spheroids after 1 and 3 days of treatment. Data represent mean ± SD from three independent experiments. * *p* < 0.05; ** *p* < 0.01. See Appendix A for complete statistical analysis.

**Figure 4 ijms-26-06957-f004:**
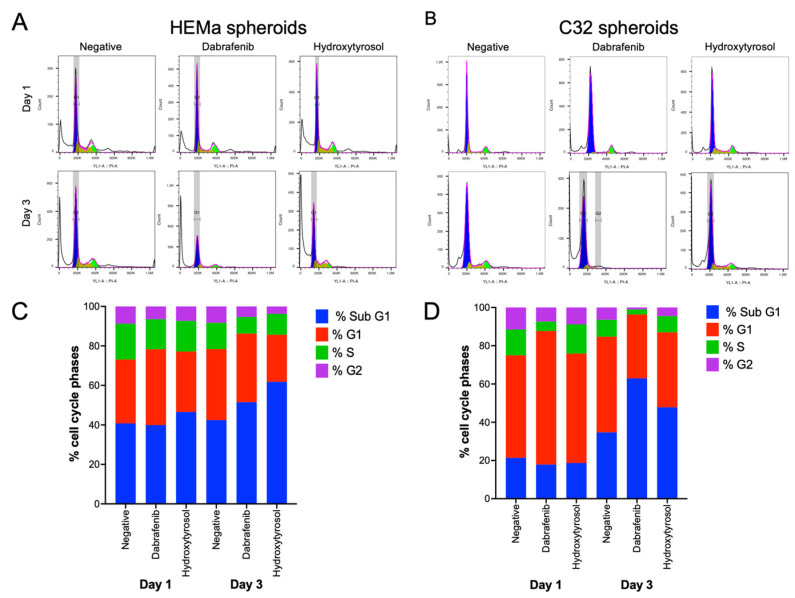
Hydroxytyrosol alters cell cycle distribution in spheroids. (**A**,**B**) Representative histograms of DNA content and quantification of cell cycle phases in HEMa (**A**) and C32 (**B**) spheroids after 1 day of treatment with HT (IC_50_), dabrafenib (IC_50_), or vehicle. (**C**,**D**) Cell cycle distribution, sub G1 phase (blue), G1 phase (red), S phase (green), and G2 phase (purple) at Day 1 and 3. Data are presented as mean ± SD from three independent experiments.

**Figure 5 ijms-26-06957-f005:**
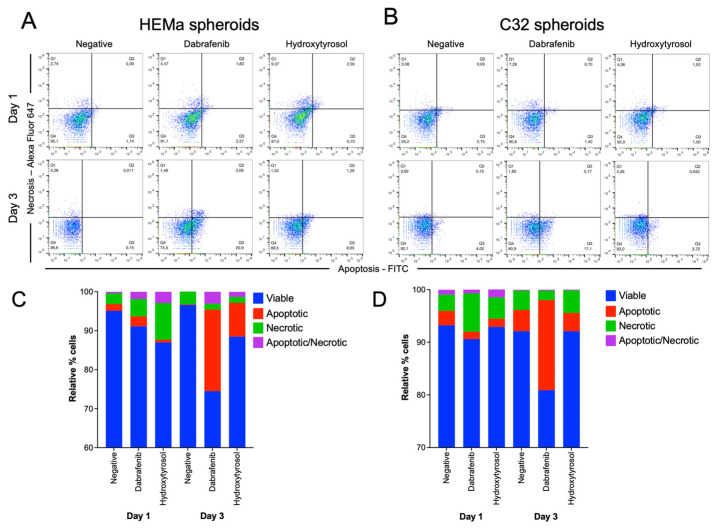
Hydroxytyrosol induces selective cell death in spheroids. (**A**,**B**) Flow cytometry plots showing necrotic (Q1), apoptotic/necrotic (Q2), and apoptotic (Q3) percentage cell markers in HEMa (**A**) and C32 (**B**) spheroids after 1 and 3 days of treatment. (**C**,**D**) Quantification of apoptotic, necrotic, and viable cell populations in HEMa (**C**) and C32 (**D**) spheroids. Data represent mean ± SD from three biological replicates.

**Figure 6 ijms-26-06957-f006:**
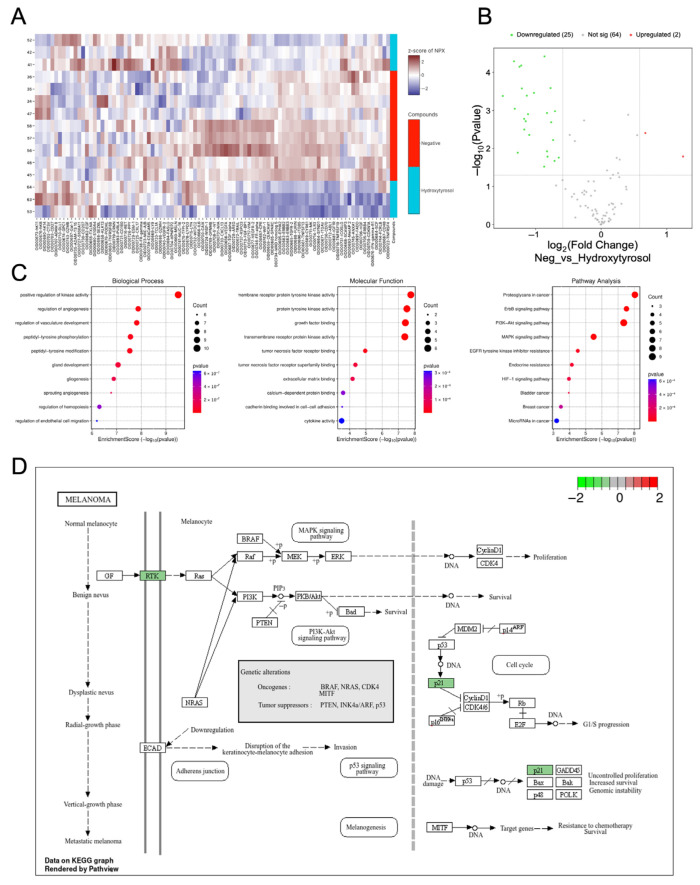
Hydroxytyrosol downregulates oncogenic signaling networks in HEMa spheroids. (**A**) Unsupervised hierarchical clustering of protein expression in HT-treated and control HEMa spheroids reveals three distinct expression patterns. (**B**) Volcano plot showing significantly downregulated (green) and upregulated (red) proteins in HT-treated spheroids versus controls. (**C**) Gene Ontology (GO) enrichment analysis of downregulated proteins. KEGG pathway analysis identifies key pathways affected by HT, including MAPK, EGFR, PI3K-Akt signaling, and (**D**) melanoma signaling pathways. Data represent mean ± SD from triplicate biological samples.

**Figure 7 ijms-26-06957-f007:**
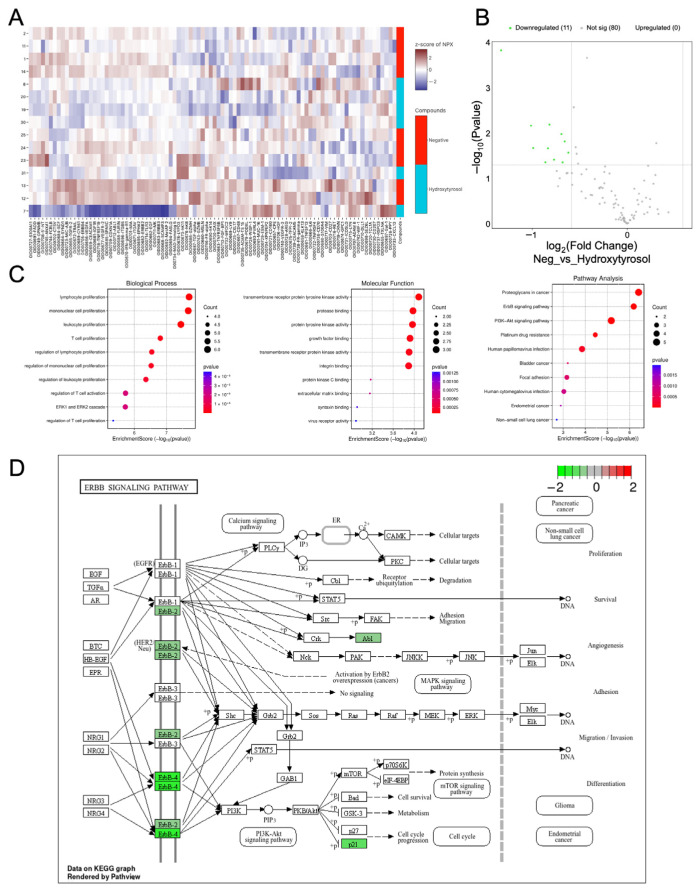
Hydroxytyrosol suppresses pro-tumor signaling in C32 melanoma spheroids. (**A**) Unsupervised clustering of protein expression profiles in C32 spheroids treated with HT or vehicle shows. (**B**) Volcano plot highlights 11 significantly downregulated proteins in HT-treated spheroids. (**C**) GO and KEGG pathway enrichment analyses reveal suppression of ERK1/2 cascade, immune responses, tyrosine kinase signaling, and proteoglycan-related pathways. (**D**) Downregulated proteins include ERBB2, ERBB4, p21, and ABL in the ERBB signaling pathways. Data are presented as mean ± SD from three biological replicates.

## Data Availability

The data presented in this study are available on request from the corresponding author.

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
