# Peer review of "Hydroxytyrosol Reprograms the Tumor Microenvironment in 3D Melanoma Models by Suppressing ERBB Family and Kinase Pathways"

_ijms, 2025, doi:10.3390/ijms26146957_

Round 1
Reviewer 1 Report
Comments and Suggestions for Authors
The authors of the study investigated the effects of hydroxytyrosol (HT), a phenolic compound derived from olive oil, on melanoma cells cultured in vitro and in 3D growth models.
The effects of HT were compared to those of the Dabrafenib inhibitor.
Specifically, the spheroid growth of tumor cells was studied, and it was found that certain concentrations of HT reduced this growth by 10-30% compared to baseline levels.
Cytotoxicity of HT was also measured in comparison to Dabrafenib, showing slightly lower toxicity than the latter.
Furthermore, the migration of spheroids was reduced in the HT-treated models compared to the control group.
Invasive capacity, assessed using a Boyden chamber, was also reduced in the HT-treated cells compared to the negative control, with an effect comparable to that achieved with Dabrafenib.
Other observed effects included the downregulation of mesenchymal EMT markers and modulation of the cell cycle, suggesting an antiproliferative effect on tumor cells.
Similar results were obtained when analyzing selective cell death rates in HT-treated tumor cells compared to untreated controls.
The manuscript is well-written and appropriately illustrated, with fluent English and clear, detailed tables and figures. The layout and presentation scheme are well-structured.
The results are clear, and the discussion is acceptable.
Overall, the study is interesting and opens up avenues for research on the effects of HT in pharmacodynamics studies.
Author Response
We thank you for taking the time to review this manuscript.
Comment 1: The authors of the study investigated the effects of hydroxytyrosol (HT), a phenolic compound derived from olive oil, on melanoma cells cultured in vitro and in 3D growth models.
The effects of HT were compared to those of the Dabrafenib inhibitor.
Specifically, the spheroid growth of tumor cells was studied, and it was found that certain concentrations of HT reduced this growth by 10-30% compared to baseline levels.
Cytotoxicity of HT was also measured in comparison to Dabrafenib, showing slightly lower toxicity than the latter.
Furthermore, the migration of spheroids was reduced in the HT-treated models compared to the control group.
Invasive capacity, assessed using a Boyden chamber, was also reduced in the HT-treated cells compared to the negative control, with an effect comparable to that achieved with Dabrafenib.
Other observed effects included the downregulation of mesenchymal EMT markers and modulation of the cell cycle, suggesting an antiproliferative effect on tumor cells.
Similar results were obtained when analyzing selective cell death rates in HT-treated tumor cells compared to untreated controls.
The manuscript is well-written and appropriately illustrated, with fluent English and clear, detailed tables and figures. The layout and presentation scheme are well-structured.
The results are clear, and the discussion is acceptable.
Overall, the study is interesting and opens up avenues for research on the effects of HT in pharmacodynamics studies.
Response 1: We sincerely thank the reviewer for the positive and thoughtful summary of our work. We appreciate the recognition of the manuscript’s clarity, organization, and relevance.
Reviewer 2 Report
Comments and Suggestions for Authors
In this study, the authors investigated the anti-tumor effects of the phenolic compound hydroxytyrosol derived from extra virgin olive oil in 3D melanoma spheroid models, using modern and robust data collection techniques.
Some points should be noted before the study is accepted in its final form.
Introduction
Lines 54-55 – confirm the term "invasive programs"
Lines 59-62 – I suggest citing the main phenolic compounds found in extra virgin olive oil, which will be discussed later
Line 65 – state the meaning of the acronym "EVOO"
Results
Lines 89-93 – I suggest the authors consider removing this section from the results section, as this information is more relevant to the methods. This way, they could begin the topic by discussing their significant results.
Page 3 - Figure 1 has very small print and may interfere with the reading and interpretation of the results. This also occurs in all figures.
Results item 2.1. Is it possible to show statistics for these results and express them in the figures? This would provide more support for the analyses, as shown in Figure 1B.
The 25 ppm dose showed no significant activity in either type of spheroid; I suggest removing this citation to improve readability.
Line 174 - Check "treatment. (Figure 3C–E)."
Line 272 - Check the topic title
Materials and methods
Lines 368–369 - Describe the concentrations of dabrafenib and ethanol used.
Author Response
Reviewer 2
We thank you for taking the time to review this manuscript.
Please find the detailed responses below and the corresponding revisions/corrections in track changes in the re-submitted files.
We hope that the revisions and changes will be appropriate to your suggestions.
In this study, the authors investigated the anti-tumor effects of the phenolic compound hydroxytyrosol derived from extra virgin olive oil in 3D melanoma spheroid models, using modern and robust data collection techniques.
Some points should be noted before the study is accepted in its final form.
Comment 1: Introduction
Lines 54-55 – confirm the term "invasive programs"
We appreciate this observation. The term "invasive programs" refers to the set of cellular and molecular mechanisms that promote tumor cell migration, invasion, and epithelial-to-mesenchymal transition (EMT), which are well-established hallmarks of malignancy. The relevant sentence has been modified to improve clarity.
“Thus, the identification of complementary or alternative therapeutic agents, particularly those that can target proliferative and invasive phenotypes, remains a high priority.”
Lines 59-62 – I suggest citing the main phenolic compounds found in extra virgin olive oil, which will be discussed later
We appreciate this helpful suggestion. In response, we have revised the sentence to explicitly list the main phenolic compounds found in extra virgin olive oil (hydroxytyrosol, tyrosol, and oleuropein), which are relevant to our study.
“Among these compounds, phenolic compounds such as hydroxytyrosol (HT), tyrosol, oleacein, oleocanthal, and oleuropein, found in extra virgin olive oil (EVOO), have emerged as promising candidates for cancer treatment and also for their diverse pharmacological properties, including antioxidant, anti-inflammatory, and anti-cancer effects”
Line 65 – state the meaning of the acronym "EVOO"
We appreciate this observation. The acronym “EVOO” (extra virgin olive oil) has been spelled out upon its first mention in the revised manuscript (Line 60).
Comment 2: Results
Lines 89-93 – I suggest the authors consider removing this section from the results section, as this information is more relevant to the methods. This way, they could begin the topic by discussing their significant results.
We appreciate this perceptive suggestion. Accordingly, we have removed the description of spheroid formation conditions (previously in Lines 89–93) and revised the Results section to focus directly on the significant findings related to HT treatment effects.
“Both non-tumorigenic (HEMa) and tumorigenic (C32) melanoma spheroids were treated from day 5 to day 11 with increasing concentrations of hydroxytyrosol (HT; 0–100 ppm). In non-tumor HEMa spheroids…”
Page 3 - Figure 1 has very small print and may interfere with the reading and interpretation of the results. This also occurs in all figures.
Thank you for pointing this out. While the figures may appear small within the compiled manuscript text, all figures have been revised to improve font size and readability. Additionally, high-resolution versions (300 dpi) of all figures have been uploaded to the submission platform to ensure optimal clarity during review and in the final publication format.
Results item 2.1. Is it possible to show statistics for these results and express them in the figures? This would provide more support for the analyses, as shown in Figure 1B.
We appreciate this important suggestion. While all detailed statistical analyses (including p-values and test types for migration and invasion assays) are provided in Supplementary Table 1, the figure legend has also been updated to reference the statistical methods and the supplementary table for full details (Lines 135-137).
“Figure 2. Hydroxytyrosol inhibits spheroid migration on Matrigel. (A) Representative images and quantification of HEMa spheroid migration over 5 days in Matrigel-coated wells. (B) C32 spheroid migration over the same period. Graphs represent spheroid outgrowth area over time, expressed as a percentage relative to Day 0. Data are shown as mean in % (n = 4 spheroids per condition) (*p < 0.05, **p < 0.01). See Supplementary Table 1 for complete statistical analysis.”
The 25 ppm dose showed no significant activity in either type of spheroid; I suggest removing this citation to improve readability.
Thank you for this observation. As suggested, we have removed the mention of the 25 ppm dose from the main Results narrative to enhance clarity and focus on the significant findings.
Line 174 - Check "treatment. (Figure 3C–E)."
Thank you for this observation. As suggested, this formatting issue has been corrected in the revised manuscript.
Line 272 - Check the topic title
Thank you for this observation. As suggested, the section title at Line 272 has been revised and changed.
“Hydroxytyrosol downregulates pro-tumorigenic protein networks in C32 spheroids”
Comment 3: Materials and methods
Lines 368–369 - Describe the concentrations of dabrafenib and ethanol used.
We agree with this comment and have now added the exact concentrations of dabrafenib and the vehicle (ethanol) used in the treatments, as described in the revised Materials and Methods section (lines 360-366).
“Spheroids were then exposed to 3-hydroxytyrosol (Sigma-Aldrich, SGM-H4291-25MG) at concentrations ranging from 1 to 100 parts per million (ppm). Dabrafenib (MedChemExpress, HY-12057) served as the positive control (BRAF inhibitor) concentration range of 0–100 ppm, and ethanol (EtOH) was used as the vehicle control, diluted to a final concentration of 0.1% (v/v). All treatments were performed in biological triplicate unless otherwise stated.”